# Informative Regions In Viral Genomes

**DOI:** 10.3390/v13061164

**Published:** 2021-06-18

**Authors:** Jaime Leonardo Moreno-Gallego, Alejandro Reyes

**Affiliations:** 1Department of Microbiome Science, Max Planck Institute for Developmental Biology, 72076 Tübingen, Germany; jaime.moreno@tuebingen.mpg.de; 2Max Planck Tandem Group in Computational Biology, Department of Biological Sciences, Universidad de los Andes, Bogotá 111711, Colombia; 3The Edison Family Center for Genome Sciences and Systems Biology, Washington University School of Medicine, Saint Louis, MO 63108, USA

**Keywords:** eukaryotic viruses, phages, orthologous gropus, random forest, ViPhOGs

## Abstract

Viruses, far from being just parasites affecting hosts’ fitness, are major players in any microbial ecosystem. In spite of their broad abundance, viruses, in particular bacteriophages, remain largely unknown since only about 20% of sequences obtained from viral community DNA surveys could be annotated by comparison with public databases. In order to shed some light into this genetic dark matter we expanded the search of orthologous groups as potential markers to viral taxonomy from bacteriophages and included eukaryotic viruses, establishing a set of 31,150 ViPhOGs (Eukaryotic Viruses and Phages Orthologous Groups). To do this, we examine the non-redundant viral diversity stored in public databases, predict proteins in genomes lacking such information, and used all annotated and predicted proteins to identify potential protein domains. The clustering of domains and unannotated regions into orthologous groups was done using cogSoft. Finally, we employed a random forest implementation to classify genomes into their taxonomy and found that the presence or absence of ViPhOGs is significantly associated with their taxonomy. Furthermore, we established a set of 1457 ViPhOGs that given their importance for the classification could be considered as markers or signatures for the different taxonomic groups defined by the ICTV at the order, family, and genus levels.

## 1. Introduction

Viruses are entities widely spread all around the biosphere. It is estimated that viral particles are 10 times more abundant than other types of microorganisms and, although their inclusion in a new life domain remains controversial, it is clear that they are not merely parasites [1,2,3]. Viruses actively participate in ecosystem remodeling, population dynamics, and a wide variety of ecological, biogeochemical, genetic, and physiological processes [4].

Despite their importance and abundance, the viral diversity has not been well characterized. Difficulties in the isolation of pure cultures and the description of viral cycles are common limitations in virus research, since less than 1% of environmental microorganisms can be grown in the laboratory [5]. Next generation sequencing techniques enabled us to partially overcome these difficulties, revolutionizing this field of virology. Deep sequencing surveys of viral communities (viromes) have revealed a diversity beyond all expectations, and they have evidenced the lack of knowledge we currently have on global viral diversity. Only about 25% of sequences in viromes from marine environments have a match (e-value ≤ 0.001) to a known sequence [6,7,8]. The large diversity and the lack of universal molecular markers make it difficult to organize and characterize known and new viral genomes.

Currently, computational tools such MetaVir2 [9], and MG-RAST [10] use available molecular databases to analyze genomes and metagenomes. Nevertheless, they are limited by considering only the fraction of the data generated that has significant similarity to previously annotated data. New approaches based not only on annotated sequence comparisons, but also on all the available information promise to be useful for analyzing individual viral genomes and viromes. For instance, Skewe-Cox et al., used protein sequence clustering to generate viral profile Hidden Markov Models (“vFams”) that were subsequently used for classifying highly divergent sequences [11]. Furthermore, the identification of highly conserved genes in specific taxonomic groups has been another approach for the taxonomic classification of viral sequences. For instance, diversity analyses of cyanophages, algae viruses, T4 and T7 phages have been conducted following this method, and they enabled the characterization of the viral diversity in the studied environments [12,13,14]. However, the mentioned studies were restricted to specific families and ecosystems.

Using another approach, Kristensen et al. constructed a collection of phage orthologous groups (“POGs”) from bacterial and archeal viruses [15,16]. Orthologous gene sets are widely used as a powerful technique in comparative genomics and for viruses it has been suggested that marker genes could be obtained using this technique [15,16]. Based on the same concept, eggNOG has been implemented in its latest version a database of orthologous groups focused on viruses (Viral OGs) [17]. However, they do not include the whole breadth of viral diversity represented in the public databases, since genomes of eukaryotic viruses were not included by either of the mentioned studies. Given the large amount of currently available genetic information, it is imperative to develop and implement new tools to reliably and efficiently analyze these data to better describe viral diversity.

Computational techniques have been used previously for similar issues in biology. Machine learning methods, for example, are algorithms which learn through the experience, attempting to classify information according to shared features. These techniques allow us to extract patterns, trends, and, finally, analyze the information using a non-deterministic way. Supervised learning algorithms such as random forest, support vector machine, and neural networks have been successfully introduced to solve complex biological problems, such as image analysis, microarray expression analysis, QTLs analysis, detection of transcription start sites, epitopes detection, protein identification and function, among others [18,19].

In this work we used the methodology proposed by Kristensen et al. for the identification of gene and domain orthologous groups from related viral sequences. We expanded the reach of this approach by incorporating genomes of eukaryotic viruses and applying random forest as a machine learning strategy to identify taxonomically informative orthologous groups. We denominated the set of orthologous groups as Eukaryotic Viruses and Phages Orthologous Groups (ViPhOGs).

## 2. Materials and Methods

### 2.1. Dataset

All viral genomes stored at the NCBI public databases in April 2015 were retrieved based on the queries previously used by Kristensen et al. [16].

To download the genomes of viruses infecting eukaryotic cells (hereafter eukaryotic viruses) available on RefSeq the query used was: *viruses[Organism] NOT cellular organisms[Organism] AND srcdb_refseq[Properties] NOT vhost bacteria[Filter] AND “complete genome”[All fields]*. The query to download the genomes of viruses infecting bacterial cells (hereafter phages) was: *viruses[Organism] NOT cellular organisms[Organism] AND srcdb_refseq[Properties] AND vhost bacteria[Filter] AND “complete genome”[All fields]*. To download complete genomes stored in the Genbank database but not in the RefSeq database, the negation of the proposition *srcdb_refseq[Properties]* was used in the queries.

Following the retrieval of viral genome sequences, a series of filtering steps was applied to the query results to remove incomplete genome sequences and to reduce the redundancy in the dataset. First, keyword depuration: definitions of entries having any of the keywords “segment”, “ORF”, ”gene”, ”mutant”, ”protein”, ”complete sequence”, “Region”, “CDS”, “UTR”, “recombinant”, or “terminal repeat” were inspected. Entries that did not correspond to complete genomes were removed. Second, genomic dereplication: genomes were clustered to a 95% sequence identity over the full length of the shorter sequence using CD-HIT-EST [20]. Only the representative sequence of each cluster, the longest one according to CD-HIT documentation, was used for further analysis. Finally, dereplication at the protein level: as in Kristensen et al. [15], to remove redundancy of sequences that are not in synteny but are essentially the same genome, genomes were clustered according to their protein content using a complete linkage approach. To identify shared proteins among genomes, all proteins from all genomes were grouped at a global sequence identity of 99% using CD-HIT. Genomes coding for 20 proteins or less must share all the proteins to be clustered while genomes coding more than 20 proteins must share at least 90% of their proteins to be clustered.

Nucleotide and protein sequences from the genomes that passed the aforementioned filters were considered as the non-redundant viral diversity available in NCBI at the time this study was conducted and were used for further analyses.

### 2.2. Gene Prediction

Genes were predicted for genomes without any annotation using Glimmer [21] as implemented in RAST-tk [22], GeneMarkS (v.2.0) [23], and Prodigal (v.2.6.3) [24]. The protein prediction was carried out separately for eukaryotic viruses and phages; and, in the particular case of GeneMark, it was possible to specify if the genome was single or double stranded according to the taxonomy annotation of each genome. Predicted proteins per genome were dereplicated using CD-HIT at 99% sequence identity to collapse the predictions made by the 3 packages.

### 2.3. Domain Prediction and ViPhOGs

To split proteins into their component domains we first used InterProScan, which combines several signature recognition methods to predict the presence of functional domains [25]. Domains were extracted and protein regions without domain annotations and comprising at least 40 residues were also kept.

In an attempt to get an annotation for proteins and protein regions without InterProScan annotations, we used them as queries against vFams. A database of hidden Markov models (HMM) built from viral RefSeq proteins [11]. Protein sequences that matched entries in the vFam database inherited the corresponding annotations, whenever these were available.

After this process, complete proteins without any domain annotation, protein regions of at least 40 residues, and domains identified by either InterProScan or vFams were considered for further analysis and referred to as viral regions from now on. Orthologous groups were built from the symmetric best matches between viral regions using the software COGsoft [26], and only considering matches with an E-value < 0.1 and that covered at least 50% of the viral region lengths. The clusters of orthologous groups built from viral regions of eukaryotic viruses and phages are, hereafter, denominated Eukaryotic Viruses and Phages Orthologous Groups (ViPhOGs).

### 2.4. Random Forest Classifiers

To test if the ViPhOGs can be used as a set of features that defines every virus or phage genome in our dataset, we aimed to correctly assign viral taxa to each genome according to the presence or absence of ViPhOGs. To solve this supervised learning task, we used the scikit-learn implementation of the random forest classifier algorithm [27] to, independently, perform the classification process at three different taxonomic levels: order, family, and genus. For each taxonomic level half of the genomes were randomly chosen for training, while the other half were used for testing the classifiers. Although randomly chosen, we constrained the selection of genomes to balance the taxa represented in both the training and testing sets. As a consequence, taxons with a single representative were not included in the classification process.

To evaluate the effect of the number of estimators in the classification we varied the number of estimators from 10 to 100 (by increments of 10 on each test), using 50 random training or testing sets for each model. The mean of the generalization score and the elapsed real time were the variables analyzed to set the optimal number of estimators in the final model.

After establishing the number of estimators for the model, we sought to reduce the number of features or ViPhOGs used for the classification. A set of ViPhOGs was pre-selected by calculating both sensitivity and precision (SP) and mutual information (MI) metrics as in Reyes 2015 [28]. The ViPhOGs selected as features were those showing both, (i) a low SP index (high sensitivity and precision for the evaluated taxa) and (ii) a high MI index for the evaluated taxa. The selected set of ViPhOGs were used for the random forest algorithm to solve the classification problems. This time, 100 random training or testing sets were used for each model.

### 2.5. Selection of Informative ViPhOGs

In order to identify the most informative ViPhOGs for each classification model, features were ranked in descending order according to their mean Gini importance. Then, each model was run again several times, but each time the number of features was reduced by $\frac{1}{5}$ of the number used in the previous run to exclude the least important ViPhOGs. This process was repeated until the model was run with the 4 most important ViPhOGs. Finally, the classification score of each iteration was plotted as a function of the number of features, and the smallest set of features that reached the highest mean classification score was selected as the set of informative ViPhOGs for each model. To depict how the viral diversity is related (or not), we built a tree using the unweighted pair group method with arithmetic mean (UPGMA tree) based on the presence or absence of informative ViPhOGs for each genome.

## 3. Results

### 3.1. The Viral Diversity Represented in Public (NCBI) Databases

We searched for all viral genomes stored in either RefSeq or Genbank databases (April 2015) and obtained 50,728 entries by using the selected set of queries (see methods). Depuration of the search results led to the exclusion of 6617 entries, as some of the keywords in their descriptions indicated that they did not correspond to complete genomes. Entries kept following the depuration step were clustered based on sequence identity in order to collapse near identical viral sequences, which resulted in an overall 57% reduction that reflects a very high redundancy in the searched databases. Bacteriophage sequences decreased from 3573 to 2071 (57.9%), while sequences of eukaryotic viruses went from 40,538 to 13,011 (32.1%). Finally, a second dereplication at the protein level was conducted following the prediction of genes for those genome accessions without protein annotations (see methods).This process led to a final reduced set of 14,057 entries, comprising 1974 bacteriophages and 12,083 eukaryotic viruses. Those accessions are considered as the non-redundant viral diversity stored in NCBI public databases (Appendix A).

According to the type of genetic material stated in the description of the accessions, these were categorized into: double stranded DNA (dsDNA), single stranded DNA (ssDNA), double stranded RNA (dsRNA), single stranded RNA —despite the sense— (ssRNA), and retro-transcribing viruses (rt-viruses). A total of 1122 accessions did not have a complete taxonomic annotation at the moment of the study; those accessions were found to be either unclassified phages (66), unclassified viruses (80), satellites (203), or assemblies from marine metagenomes (773). The longest genome belonged to the dsDNA virus *Pandoravirus salinus* (NC_022098) with a genome length of 2,473,870 bp, whereas the smallest genome belongs to the ssRNA *Lucerne transient streak satellite virus* with 324 bp. In general, DNA viruses are larger than RNA viruses (Mann-Whitney test: *p*-value = 1.12 × 10${}^{-135}$). A comparison of the genome length distribution of phages and eukaryotic viruses shows that dsDNA and ssDNA phages tend to have larger genomes than dsDNA and ssDNA eukaryotic viruses (Mann-Whitney test when comparing dsDNA viruses: *p*-value = 7.19 × 10${}^{-6}$; Mann-Whitney test when comparing ssDNA viruses: *p*-value = 3.08 × 10${}^{-64}$). In the case of ssRNA viruses, eukaryotic viruses tend to have larger genomes than phages (Mann-Whitney test when comparing ssRNA viruses: *p*-value = 7.89 × 10${}^{-16}$) (Figure 1).

The set of 14,057 non-redundant genomes code for a total of 442,007 proteins, where we observe that the number of genes is directly proportional to the length of the genome. Interestingly, a linear regression suggests a gene density of 12 proteins per kilobase in the case of phages, while in the case of eukaryotic viruses the gene density is only about 2.5 proteins per kilobase; indicating a lower gene density for eukaryotic viruses in comparison with phages (Figure 2).

### 3.2. Eukaryotic Viruses and Phages Orthologous Groups (ViPhOG)

We searched for domains in all identified and predicted proteins using InterProScan [25]. Domains were found only in 52.59% of the proteins (232,033 proteins), which means that even for the sequences stored in public databases half of the information belongs to the viral dark matter. In an attempt to gain further information, unannotated proteins were used as a query against vFams, but only 39,344 (8.9%) proteins had a significant match to entries in this database.

Given the proportion of unannotated sequences, protein regions without domain or vFam annotation were also considered for further analysis, meaning that the final set of viral regions consisted of: 365,368 annotated regions (309,251 InterPro domains + 56,137 vFam matches), 157,591 unannotated regions of at least 40 residues (69,333 regions in between annotated domains + 88,258 regions in between vFam matches) and 170,637 unannotated proteins (proteins with no hit to vFam or InterProScan). The set of orthologous groups was built from the symmetric best matches between viral regions using the software COGsoft [4] (see methods). A total of 31,150 ViPhOGs with at least three members were obtained. Interestingly, most of the ViPhOGs were built from a single type of viral regions: unannotated proteins (9953), unannotated regions (8103) or annotated regions (8023). Among all possible combinations of viral region types, the highest number of clusters was obtained for the combination of unannotated proteins and unannotated regions (2309) (Figure 3). This suggests that although the vast majority of regions and proteins in viral genomes are uncharacterized, they are conserved among the different chosen viruses.

The median amount of regions clustered in a ViPhOG was 5 (IQR:3,11) with the largest ViPhOG having 3440 regions from 1180 different genomes of both phages and eukaryotic viruses. This large ViPhOG contained regions mainly annotated as Helicases. However, it was not the only ViPhOG that comprised a rather large number of regions, as a total of 1081 ViPhOGs contained more than 100 regions (Appendix A). In terms of the host type, we found 14,746 ViPhOGs represented exclusively by eukaryotic viral genomes, 10,100 ViPhOGs represented only by phage genomes and the remaining 6304 ViPhOGs were represented by both phages and eukaryotic viral genomes. As a ViPhOG may include paralogs, any given genome can contribute with several regions to a single ViPhOG. However, the number of regions per genome for each ViPhOG was on average 1.008 (max: 9.162), which indicates that the vast majority of ViPhOGs are composed of orthologs instead of paralogs. This is also evidenced in Appendix A, where most of the ViPhOGs have the same number of genomes as regions, indicating that each genome contributed only one region to each orthologous group.

### 3.3. A Random Forest Classifier Correctly Classifies Viral Genomes According to the Presence of ViPhOGs

We used the Scikit-learn random forest implementation to test if ViPhOGs can be used as features to predict taxonomy (see methods). From the model testing, it was determined that in general a model with 60 estimators had a reasonably good balance between classification-score and computation time, as models with 60 estimators result in a high classification score and less variance (Appendix A). Therefore, a random forest classifier with 60 estimators was run separately for each of the evaluated taxonomic levels. For each taxonomic level, all genomes classified into a taxon at the analyzed level were used. For the order level, the algorithm received a matrix of 1031 ViPhOGs and 4698 genomes to classify into 7 Orders. In the case of family, the matrix contained 11,328 ViPhOGs and 11,978 genomes from 84 different families, and for the genus level, the size of the matrix was 20,310 ViPhOGs and 10,151 genomes from 335 different genera. For each case, matrices were split in 100 train and test (70:30 distribution) sets. The mean accuracy score achieved was 99.06%, 95.60%, and 89.58% for order, family, and genus, respectively (Figure 4, Appendix A).

As the classification score suggests, the chosen algorithm excelled at accurately classifying the genomes into their respective taxonomic groups; where most of the misclassification cases were a small proportion of the genomes represented by each of the assessed taxons (Appendix A). The 10 most common classification mistakes per taxonomic level are shown in the Appendix A. Although we observed that the classification error does not perfectly correlate with the total number of available genomes for the classification, it is evident that the lower the number of genomes available for a given taxon, the higher the classification error (Appendix A).

### 3.4. Informative ViPhOGs: Signatures of the Taxonomy of Viral Genomes

As the good performance of the models suggests, there is a set of ViPhOGs that allows to identify with high accuracy the taxonomic group of each genome. Therefore, we aimed at minimizing the number of ViPhOGs capable of reaching high accuracy of classification for each given taxonomic level (see methods). That approach allowed us to determine that a reduced set of 20 ViPhOGs was enough to achieve a high accuracy score for the order level. For the family and genus levels, the number of ViPhOGs needed was 388 and 1392, respectively (Appendix A). We designated those ViPhOGs as “Informative ViPhOGs”. Their taxonomic assignment and their functional annotation (if available) is presented in the Appendix A.

We used the set of informative ViPhOGs to build an UPGMA tree that delineates the viral diversity clustering, based on the presence or absence of these genomic characteristics (see methods). Genomes that belong to a defined order constitute a defined clade in the tree. Furthermore, in all cases except for *Caudovirales*, there was a consistent branching of the orders containing the corresponding families and genera designated by the International Committee on Taxonomy of Viruses (ICTV) (Figure 5). A closer look at the tree revealed some interesting viral features.

(i) Bacteriophages. The family *Tectiviridae* (dsDNA viruses) shares a clade with the genus Rosemblanvirus and Salasvirus, both members of the family *Podoviridae* (also dsDNA viruses from the order Caudovirales), while the other bacteriophage families *Inoviridae*, *Microviridae* (both ssDNA), and *Leviviridae* (ssRNA) appear as independent clades without shared characteristics among them or members of the order *Caudovirales*. Regarding archaeal viruses, the family *Fuselloviridae* and the order *Ligamenvirales*, which includes the families *Rudiviridae* and *Lipothrixviridae*, form a single clade. Furthermore, most of the families of archaeal viruses had very few representatives, revealing a bias in the explored diversity;

(ii) Characteristics shared between eukaryotic viruses and phages. The order *Herpesvirales* appears as a sister clade of a subset of the *Caudovirales*, in particular, members of the *Myoviridae* family. Moreover, the Nucleo-Cytoplastmatic Large DNA Viruses (NCLDV) group is enclosed by the *Caudovirales* clade. We looked for ViPhOGs present in members of all NCLDV families and found ViPhOGs number 937 and 1598, which are associated with helicase domains and, as mentioned before, prevalent among dsDNA viruses in general; ViPhOG 821, which codes for a ribonucleotide reductase and is shared mainly among members of the families *Myoviridae*, *Herpesviridae*, and *Poxviridae*; ViPhOG 865, a serine/threonine kinase domain, and ViPhOG 72, an EF binding domain, both also very common in members of *Herpesviridae*;

(iii) RNA viruses. Those from the family *Chrysoviridae* (dsRNA) grouped with *Totiviridae* (dsRNA), in particular with the genus *Totivirus* that also infects fungi. Interestingly, there was a clade formed by different positive sense ssRNA viruses, which had no other common taxonomic assignment. This clade included the families *Tombusviridae*, *Nodaviridae*, *Bromoviridae*, *Virgaviridae*, *Togaviridae*, *Hepeviridae*, *Closteroviridae*, and the families of the order *Tymovirales*.

## 4. Discussion

In recent years, as metagenomics has revealed a great diversity of phages from different biomes, and evidenced the huge unexplored diversity that the viral world holds. The use of genetic signatures to describe and characterize the diversity of specific groups of viruses have been successfully applied in diverse contexts [29,30]. Different strategies have been described including POGs [15,16], vFams [11], viralOGs from EggNOG [31], and pVOGs [32]. Here, we followed the methodology by Kristensen et al., and took it a step further by extending the search of orthologous groups beyond bacterial and archeal viruses to also include eukaryotic viruses, which consolidated a final set of 14,057 non-redundant genomes.

We took a non-waste-information approach in order to get orthologous groups among (i) annotated domains, (ii) unannotated regions of annotated proteins, and (iii) unannotated proteins, all derived from the non-redundant diversity of viruses stored in public databases. This strategy proved to be useful given that a large majority of the ViPhOGs were constituted solely or in combination of unannotated regions or unannotated proteins. Finally, we established a comprehensive set of 31,150 orthologous groups that we denominated ViPhOGs.

As the ICTV provides a single classification scheme that reflects the evolutionary relationship among viruses, we evaluated the possibility that the presence or absence of ViPhOGs in viral genomes reflected the ICTV taxonomy using a machine learning approach. The low misclassification scores reached by the random forest algorithm suggested that the use of ViPhOGs as features for performing taxonomic classification of viruses had great potential. Therefore, we determined the subset of informative ViPhOGs that could be considered as markers or signatures for the different taxonomic groups defined by the ICTV at the order, family, and genus levels.

We found a high degree of agreement between clustering identified using informative ViPhOGs and the monophyletic orders described by ICTV. Example of those were the *Nidovirales* [33], *Ligamenvirales* [34], *Mononegavirales* [35], and *Tymovirales* [36] whose branches shows a clear separation in accordance to the proposed families and genera.

Importantly, the current approach has been consistent with recent changes in the ICTV taxonomy. For example, the family *Pneumoviridae*, whose members were considered as a subfamily of the family *Paramyxoviridae* up till 2016 [35,37]. Our classification mechanism used the ICTV classification from 2014, but was capable of showing the separation of the family *Paramyxoviridae*. The tree clearly showed how eukaryotic viruses from the genera *Metapneumovirus* and *Pneumovirus* (now known as *Metapneumovirus* and *Orthopneumovirus*, respectively) form a separate clade (now Family Pneumoviridae), whose sister clade is the family *Paramyxoviridae*.

Despite the absence of a link between several ssRNA(+) families in the ICTV taxonomy of 2014, in the ViPhOG-based tree built here families *Tombusviridae*, *Nodaviridae*, *Bromoviridae*, *Virgaviridae*, *Closteroviridae*, and *Hepeviridae* were grouped together with the families of the order *Tymovirales*. In the most recent ICTV taxonomy the families *Bromoviridae*, *Virgaviridae*, *Togaviridae*, and *Closteroviridae* were assigned to the order *Martellivirales*; and the family *Hepesviridae* to the order *Hepelivirales*. These two new orders (*Hepelivirales* and *Martellivirales*), together with the *Tymovirales*, belong now to the Class *Alsuviricetes* and the phylum *Kitrinoviricota*. Furthermore, in our tree the families *Tombusviridae* and *Nodaviridae* are a sister clade of what seems now to be the *Alsuviricetes* clade. *Tombusviridae* and *Nodaviridae* belong now to the orders *Tolivirales* (Class: *Tolucaviricetes*) and Nodamuvirales (class: *Magsaviricetes*), respectively. All classes, together with *Alsuviricetes*, constitute now the Kingdom *Kitrinoviricota*. This suggests that a tree generated with conserved amino acid features could identify basal evolutionary relationships among viruses matching the new scope of the ICTV [38].

Misclassification cases were very limited and more common in the lowest taxonomic level (Genus) than in the highest taxonomic level (Order). Although we did not observe a perfect negative correlation between the number of genomes available and the number of misclassification cases, we did observe that genera like *Mupapillomavirus*, *Yetapoxvirus*, *Kappapapillomavirus*, and families such as *Alphatetraviridae* and *Amalgaviridae* with two or three genomes available per taxa were frequently misclassified. Another kind of misclassification event occurred between related taxa. Such was the case for eukaryotic viruses of the Genus *Vesiculovirus* that were confounded with members of the Genus *Sprivivirus*. Both genera belong to the family *Rhabdoviridae*. Interestingly, this pair of genera share more ViPhOGs between each other than against any other member of the family *Rhabdoviridae*. This observation could be the basis of a more in-depth study which could potentially lead to the suggestion of both genera being part of a new sub-family which separates them from the rest of the family. Lastly, regarding misclassification events, we want to acknowledge that there is still a place for improvement of the classification models. We identified misclassification cases where a taxon was misclassified and the confusion does not appear to be directed by genomic relatedness. As an example we chose to discuss the case of the *Caulimoviridae* family. This family had 123 representative genomes in our database and in 10% of the cases it was misclassified as *Myoviridae*. Only a few representative genomes of each family have (at most) 3 ViPhOGs in common (ViPhOGs number 731, 1158, and 269). Those 3 ViPhOGs are not informative ViPhOGs for *Myoviridae*, and appear to be present in several different viral families and clades, therefore, there is no clear answer to why the classifier confused these two unrelated families.

One of the major strengths of the presented work is that, in addition to genomes of prokaryotic and archeal viruses, we included genomes of eukaryotic viruses. As expected, not a single ViPhOG was present in all viral genomes. Viruses do not encode for ribosomes or any other universal markers that allow the study of their phylogenetic relationships. Furthermore, it has been accepted that viruses have not evolved from a single common ancestor [3,39,40,41], which might be reflected in the high number of polytomies observed in the informative ViPhOGs tree. Besides the absence of an universal ViPhOG, a not negligible number of ViPhOGs were formed by regions from phages and eukaryotic viruses. Further analyses would be needed to determine if the fact that a ViPhOG is shared between eukaryotic and prokaryotic/archeal viruses is due to functional convergence, or if it is because those viruses presumably have an evolutionary relationship as is the case for *Herpesviridae* and *Siphoviridae* [42,43,44] or as ssRNA(+) viruses, which presumably co-evolved with their hosts before they split into eukaryotes [3,45].

The fact that a machine learning approach, based solely on genomic features reached a high score when classifying viruses in their assigned taxa, highlights how the viral taxonomy based on ecological (e.g., pathogenicity and host range) and molecular (e.g., composition of the virus genome and sequence similarity) features is a robust system able to depict the evolutionary relationships among viruses. The informative ViPhOGs dataset is, therefore, nothing but a reflection of the efforts done to establish a taxonomic system for viruses and the strength of machine learning algorithms that were able to depict patterns among a comprehensive dataset. We consider that the result, the ViPhOGs and the informative ViPhOGs datasets, may be used as a start point to hypothesize about the genetic relationships among known viral groups and as a useful tool to attempt to characterize and define the viral dark matter that is being exposed via metagenomics. We released the ViPhOGs dataset hoping that: (i) the community can use it as a tool to explore the genetic relationships among viral clades encouraging viral research, (ii) to facilitate the exploration of specific viral groups by the use of its ViPhOGs, and (iii) to obtain viral profiles in specific biomes. We want to encourage the community to exploit the benefits of the use of this comprehensive set of orthologous groups in a world of fast evolving entities that quickly lose their protein sequence conservation.

## Figures and Tables

**Figure 1 viruses-13-01164-f001:**
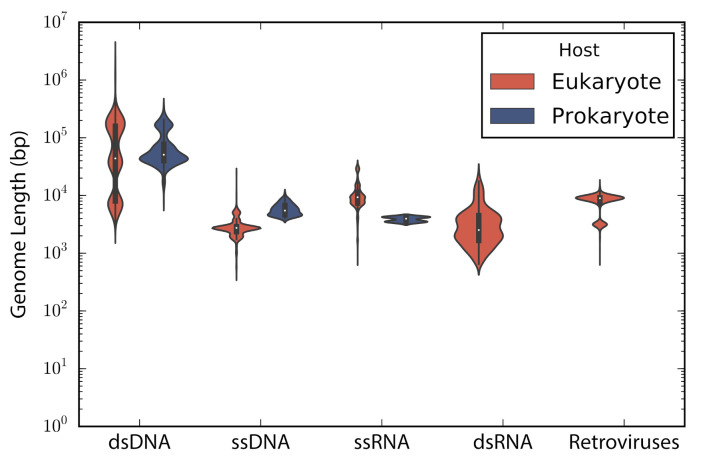
Genome length distribution. Violin plots show the genome length distribution of the non-redundant viruses used in the study. Viruses are grouped by the type of genetic material and type of host. The inner boxplot shows the median (white circle) and interquartile range (whisker plots).

**Figure 2 viruses-13-01164-f002:**
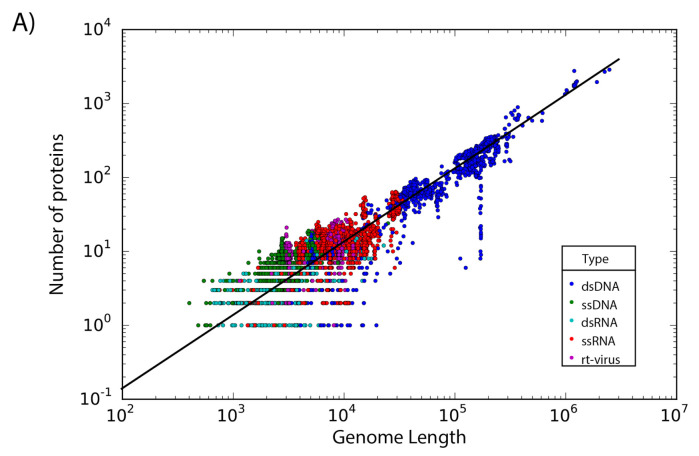
Gene density. Scatter plot showing the gene density of (**A**) eukaryotic viruses and (**B**) phages. Viruses are colored by the type of genetic material and each dot represents a genome. Best-fit lines with 95% confidence intervals from linear regression are plotted.

**Figure 3 viruses-13-01164-f003:**
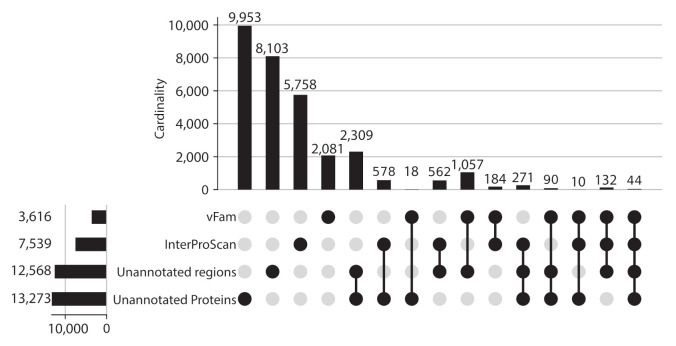
ViPhOGs composition: Most of the orthologous regions are made of unknown regions. Plot showing the ViPhOGs composition according to the type of region it has (vFam, InterProScan, unannotated region, unannotated protein). The horizontal bar plot represents the number of ViPhOGs that possess at least one region of the indicated type. Filled dots indicate which combination of types is being considered and the vertical bar plot shows the number of ViPhOGs for each combination.

**Figure 4 viruses-13-01164-f004:**
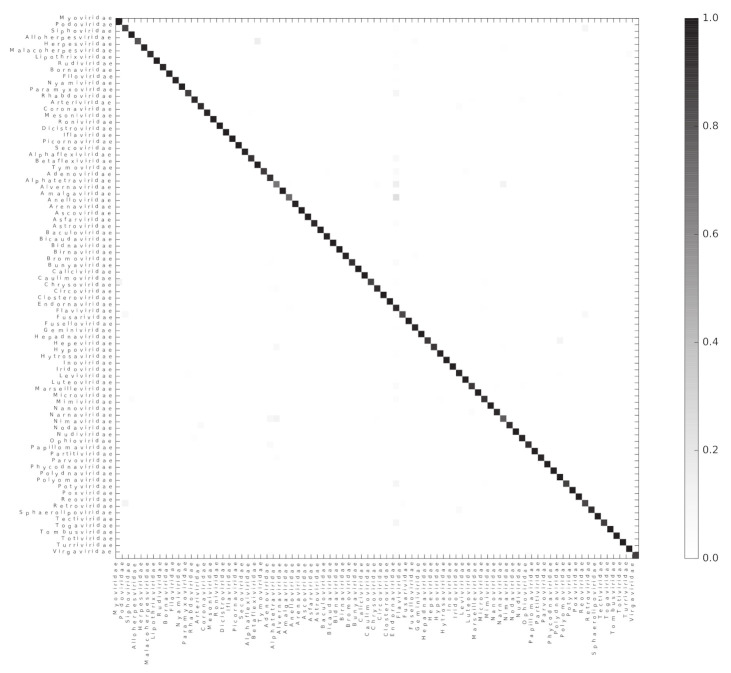
Random forest accurately classifies genomes into their respective families. A heatmap representing the confusion matrix obtained after classifying viral genomes at the family level. The color code indicates the proportion of genomes of the Family in the x-axis classified as a genome of the Family in the y-axis.

**Figure 5 viruses-13-01164-f005:**
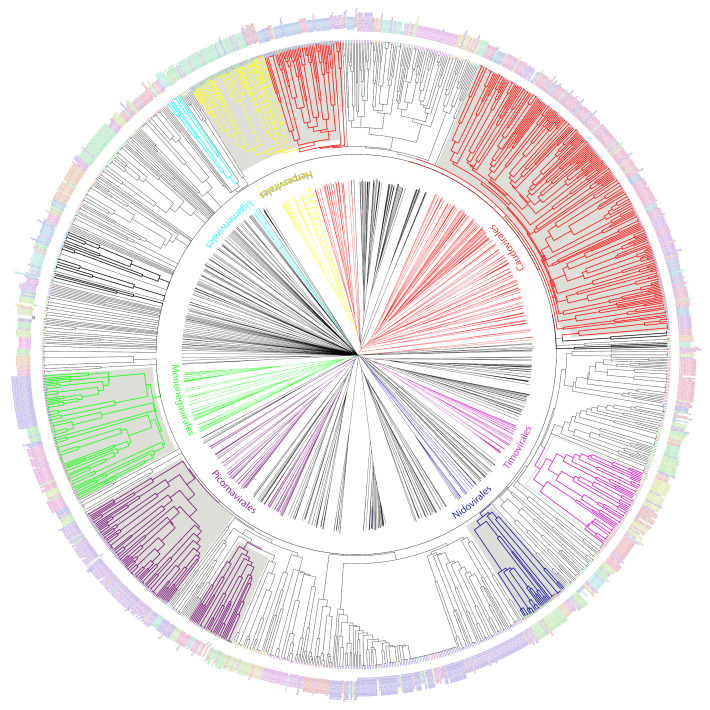
UPGMA tree representation of the non-redundant viral diversity. Unrooted (center) and circular middle point rooted (outer circle) representations of an UPGMA tree of the non-redundant viral diversity, based on the presence or absence of informative ViPhOGs. Colored branches highlight ICTV designated Orders. Bold black branches highlight phage families without an order assignment. Names between the trees indicate the name of the ICTV taxonomic order colored in the same color for the branches. Tip labels indicate the family of each genome and are colored to facilitate their differentiation within an order not to provide a different color to each family.

## Data Availability

All genome accessions are provided in Appendix A. All the scripts used to analyze and process the data are available in a github repository. Both, the ViPhOGs and informative ViPhOGs sets are available at the Open Science Framework (OSF) ViPhOGs project.

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
