# Peer review of "Informative Regions In Viral Genomes"

_viruses, 2021, doi:10.3390/v13061164_

Round 1

Reviewer 1 Report

In "Informative Regions in Viral Genomes" the authors expand on current computational methods for assessing viral sequences for similarities to develop a novel method that breaks viral genomes into similarity groups termed ViPhOGs. Using this method, they have demonstrated an ability to accurately, taxonomically classify viral genomes. Overall, this work is thoroughly depicted and described in a manner that is accessible to non-computational researchers as well. I have a few questions/suggestions for the authors.

  1. Page 3, line 90 -- In culling the dataset, entries with "complete sequence" were removed. Would this not result in full viral sequences that could be useful being removed as well? I'm unclear as to why this particular phrase was selected in the depuration process.
  2. Page 4, line 160 -- change "runned" to "run"
  3. Page 4, Line 164 -- please define the acronym UPGMA
  4. Discussion -- while the authors have certainly covered the usability of this database set for researchers examining similarities/differences between viral taxons, it would be interesting to know if the authors saw a potential use of these sequences for better characterization of environmental viromes including not only classification of viruses within these communities, but the use of these sequences for quantitation in response to environmental changes or even possibly as probes for tracking specific viruses.

Author Response

  1. Page 3, line 90 -- In culling the dataset, entries with "complete sequence" were removed. Would this not result in full viral sequences that could be useful being removed as well? I'm unclear as to why this particular phrase was selected in the depuration process.

Answer. You are right. Many complete genomes of segmented viruses include the keywords “complete sequence” (and “segment”). What we did was to examine the definition of entries that contain those keywords. Entries that did not correspond to complete genomes were removed (e.g. complete genes). You can see in Table S1 that many of the chosen entries include the word “complete sequence” and were chosen as representatives of the non-redundant diversity of viruses. We have rephrased the paragraph to avoid confusion. It now reads:

 “definitions of entries having any of the keywords “segment”, “ORF”, ”gene”, ”mutant”, ”protein”, ”complete sequence”, “Region”, “CDS”, “UTR”, “recombinant” or “terminal repeat” were inspected. Entries that did not correspond to complete genomes were removed.”

  1. Page 4, line 160 -- change "runned" to "run"

Answer. Changed.

  1. Page 4, Line 164 -- please define the acronym UPGMA

Answer. The definition of UPGMA is now included.

  1. Discussion -- while the authors have certainly covered the usability of this database set for researchers examining similarities/differences between viral taxons, it would be interesting to know if the authors saw a potential use of these sequences for better characterization of environmental viromes including not only classification of viruses within these communities, but the use of these sequences for quantitation in response to environmental changes or even possibly as probes for tracking specific viruses.

Answer. We believe that a future extension of the ViPhOGs could be of great impact in the characterization of environmental viromes. Currently it could be used if the interest is to identify viruses closely related to well characterized viruses, as those selected in this study that are present in Genbank and public databases. In order to explore the larger diversity of environmental viromes, a need of creating ViPhOGs from conserved hypothetical proteins derived from viral genomes assembled from environmental viromes will be ideal. As for the tracking of specific viruses, we have mentioned in our discussion that these ViPhOGs could help “… (ii) to facilitate the exploration of specific viral groups by the use of its ViPhOGs, and (iii) to obtain viral profiles in specific biomes.”

Reviewer 2 Report

The authors presented a very interesting work based on bioinformatic analysis. I have no comments, although perhaps my qualifications do not allow me to find any errors in the work performed. It seems to me that the manuscript can be published in its present form.

I only have a few small comments.

  1. In the abstract, you should first provide a transcript for ViPhOGs.
  2. Formally, I do not like the division into viruses and phages, since phages are also viruses, and in this aspect, the authors should compare the whole (all viruses) with phages (only bacteriophages), but in the manuscript eukaryotic viruses and bacteriophages are characterized. Maybe it is better either to introduce a different abbreviation for eukaryotic viruses and phages, or in the abstract indicate more clearly this feature of the present abbreviation
  3. line 225. I would replace the word "used" with something else, for example, "chosen" or "selected", since formally the authors did not use viruses, but information about them

Author Response

  1. In the abstract, you should first provide a transcript for ViPhOGs.

Answer. A definition for ViPhOGs is now included.

  1. Formally, I do not like the division into viruses and phages, since phages are also viruses, and in this aspect, the authors should compare the whole (all viruses) with phages (only bacteriophages), but in the manuscript, eukaryotic viruses and bacteriophages are characterized. Maybe it is better either to introduce a different abbreviation for eukaryotic viruses and phages, or in the abstract indicate more clearly this feature of the present abbreviation

Answer. Thanks for this appreciation. It really seems odd to talk about viruses and phages. To avoid confusions we removed the abbreviation. Now it is stated “eukaryotic viruses” every time we want to refer to eukaryotic viruses, phages when we only want to refer to bacteriophages, and viruses when we want to talk about the whole set (all viruses). Also, it is stated that the definition for ViPhOGs is Eukaryotic Viruses and Phages Orthologous groups. This is highlighted due to the previous existence of VOGs (Viral Orthologous Groups, which is only for Eukaryotic viruses), while the POGs is Phage Orthologous Groups and only for bacteriophages. Given that here we include both, eukaryotic and prokaryotic viruses we wanted this to be reflected in the name.

  1. line 225. I would replace the word "used" with something else, for example, "chosen" or "selected", since formally the authors did not use viruses, but information about them

Answer. We changed it for “chosen” the sentence now reads “This suggests that although the vast majority of regions and proteins in viral genomes are uncharacterized, they are conserved among the different chosen viruses.”